# Antibiotics in Necrotizing Soft Tissue Infections

**DOI:** 10.3390/antibiotics10091104

**Published:** 2021-09-13

**Authors:** Tomas Urbina, Keyvan Razazi, Clément Ourghanlian, Paul-Louis Woerther, Olivier Chosidow, Raphaël Lepeule, Nicolas de Prost

**Affiliations:** 1Médecine Intensive Réanimation, Hôpital Saint-Antoine, Assistance Publique-Hôpitaux de Paris (AP-HP), 75571 Paris, France; tomas.urbina@aphp.fr; 2Sorbonne Université, Université Pierre-et-Marie Curie, 75001 Paris, France; 3Médecine Intensive Réanimation, Hôpitaux Universitaires Henri Mondor-Albert Chenevier, Assistance Publique-Hôpitaux de Paris (AP-HP), 94010 Créteil, France; keyvan.razazi@aphp.fr; 4Groupe de Recherche Clinique CARMAS, Faculté de Médecine, Université Paris Est Créteil, 94010 Créteil, France; 5Service de Pharmacie, Hôpitaux Universitaires Henri Mondor-Albert Chenevier, Assistance Publique-Hôpitaux de Paris (AP-HP), 94010 Créteil, France; clement.ourghanlian@aphp.fr; 6Unité Transversale de Traitement des Infections, Département de Prévention, Diagnostic et Traitement des Infections, Hôpitaux Universitaires Henri Mondor-Albert Chenevier, Assistance Publique-Hôpitaux de Paris (AP-HP), 94010 Créteil, France; raphael.lepeule@aphp.fr; 7Département de Prévention, Diagnostic et Traitement des Infections, Hôpitaux Universitaires Henri Mondor-Albert Chenevier, Assistance Publique-Hôpitaux de Paris (AP-HP), 94010 Créteil, France; paul-louis.woerther@aphp.fr; 8Research Group Dynamic, Faculté de Santé de Créteil, Université Paris-Est Créteil Val de Marne (UPEC), 94010 Créteil, France; olivier.chosidow@aphp.fr; 9Service de Dermatologie, Hôpitaux Universitaires Henri Mondor-Albert Chenevier, Assistance Publique-Hôpitaux de Paris (AP-HP), 94010 Créteil, France

**Keywords:** necrotizing soft tissue infections, antibiotic, pharmacokinetics, pharmacodynamics, tissue diffusion, anti-toxinic, piperacillin-tazobactam, clindamycin, beta-lactam

## Abstract

Necrotizing soft tissue infections (NSTIs) are rare life-threatening bacterial infections characterized by an extensive necrosis of skin and subcutaneous tissues. Initial urgent management of NSTIs relies on broad-spectrum antibiotic therapy, rapid surgical debridement of all infected tissues and, when present, treatment of associated organ failures in the intensive care unit. Antibiotic therapy for NSTI patients faces several challenges and should (1) carry broad-spectrum activity against gram-positive and gram-negative pathogens because of frequent polymicrobial infections, considering extended coverage for multidrug resistance in selected cases. In practice, a broad-spectrum beta-lactam antibiotic (e.g., piperacillin-tazobactam) is the mainstay of empirical therapy; (2) decrease toxin production, typically using a clindamycin combination, mainly in proven or suspected group A streptococcus infections; and (3) achieve the best possible tissue diffusion with regards to impaired regional perfusion, tissue necrosis, and pharmacokinetic and pharmacodynamic alterations. The best duration of antibiotic treatment has not been well established and is generally comprised between 7 and 15 days. This article reviews the currently available knowledge regarding antibiotic use in NSTIs.

## 1. Introduction

Necrotizing soft tissue infections (NSTIs) are rare life-threatening bacterial infections characterized by an extensive necrosis of skin and subcutaneous tissues. NSTIs can affect any part of the body but the extremities—particularly the lower limbs—are most frequently involved [1,2,3,4]. Most patients developing NSTIs have previous comorbidities, including diabetes mellitus, obesity, cardiovascular disease, intravenous drug use, and immunosuppression [1,2,3]. Infection can spread after traumatic injuries, minor breaches of the skin or mucosa, and even non-penetrating soft tissue injuries [1]. Mortality ranges from 10 to 30% according to initial patient severity, and morbidity among survivors includes potential amputations and profound impact on long-term health-related quality of life [5,6,7]. Initial urgent management of NSTIs relies on broad-spectrum antibiotic therapy, rapid surgical debridement of all infected tissues and, when present, treatment of associated organ failures in the intensive care unit. The time to surgery is one of the main modifiable prognostic factors with, in a recent meta-analysis, a significantly lower mortality rate when surgery was performed within six hours of hospital admission [8]. High-volume centers, caring for at least three patients per year, may also contribute to improving prognosis [9].

As a consequence of their rarity, data on optimal antibiotic treatment in NSTI are scarce [10] and current guidelines [11,12,13,14] are mainly derived from observational studies and experimental data. Antibiotic therapy for NSTI patients faces several challenges and should ideally achieve the following goals: (1) carry broad-spectrum activity against gram-positive and gram-negative pathogens because of frequent polymicrobial infections, considering extended coverage for multidrug resistance in selected cases; (2) decrease toxin production, mainly in proven or suspected group A streptococcus (GAS) infections; and (3) achieve the best possible tissular diffusion in the face of impaired regional perfusion, tissue necrosis, and pharmacokinetic and pharmacodynamic alterations among these frequently critically-ill patients.

The aim of this article is to review the currently available knowledge regarding antibiotic use in NSTIs and provide clinicians with a practical tool to use at bedside.

## 2. Microbiology of NSTIs

Causative organisms vary widely according to infection site, underlying conditions, but also from one region of the world to the other [4,15,16,17,18]. In most cases, the infection is polymicrobial (so-called type I infections), involving gram-positive cocci, *Enterobacteriaceae*, nonfermenting bacilli and anaerobic bacteria. Anaerobic, aerobic, and facultative anaerobic bacteria act synergistically and fuel a cycle of bacterial colonization and inflammatory tissue necrosis [19]. However, approximately one third of NSTIs are monomicrobial (type II infections), involving mainly GAS and *Staphylococcus aureus* [4,7,19,20]. Although predominant in monomicrobial and upper-extremity infections, GAS can be documented in up to 40% of NSTIs overall [21]. A recent study has highlighted that streptococcal NSTIs were associated with different host responses than polymicrobial NSTIs, with a higher expression of interferon-inducible mediators and lower expression of extracellular matrix components [19]. In some series, *S. aureus* was reported to be more prevalent than GAS in monomicrobial NSTIs [18]. Community-associated methicillin-resistant *S. aureus* carrying the Panton-Valentine leucocidin was reported in the United States [22,23]. Classic clostridial gangrenes have become rare, while the incidence of gram-negative infections and multi-drug resistant organisms is increasing [21,24]. The most commonly isolated microorganisms are summarized in Table 1.

## 3. Microbial Documentation of NSTIs: A Challenge for Microbiologists

Because of their uncertain contribution to the microbiological diagnosis, superficial samples should be avoided [26]. Contrariwise, deep samples collected at the interface between healthy and necrotized tissues by the surgeon during initial debridement and blood cultures are paramount, allowing for the identification of causative pathogens in approximatively 90% of cases [17,19,27,28]. As for any other deep suppuration, surgical samples are affixed on slides and Gram stained for extemporaneous microscopic direct examination. The detection of yeasts or the visualization of staphylococci, possibly leading to the use of rapid molecular tools aiming to detect methicillin resistant *Staphylococcus aureus* (MRSA), can have an immediate impact on antimicrobial treatment. As recommended by the European Society of Clinical Microbiology and Infectious Diseases (ESCMID) [29], samples are seeded on solid and liquid media and incubated for up to 5 days, in order to grow all the present bacteria, including anaerobes and difficult-to-grow bacteria. Because of the deep nature of the studied samples and the severity of NSTI, all growing bacteria of medical interest should be identified, and their antibiotic susceptibility tested. Nevertheless, as presented in Table 1, pathogenecity of different micro-organisms is still a matter of debate [25,30].

The emergence of next-generation sequencing (NGS) methods is transforming medical diagnosis, including in infectious diseases. These methods include targeted metagenomics in which a PCR step (aiming to restrict the panel of microorganisms searched similarly to the 16S metagenomics for bacteria) precedes the sequencing, and shotgun metagenomics, a method consisting in sequencing the whole extracted nucleic acids (DNA and/or RNA). In the latter case, after subtraction of human sequences, analysis of the remaining reads leads to the identification of all the microorganisms present, including bacteria, viruses, yeasts, or parasites. Like in many other situations [31,32,33], shotgun metagenomics has demonstrated its better ability to detect a broad range of pathogens, particularly strict anaerobes compared to other methods [34]. Interest for this method is enhanced by the recent description of a complex pathobiome in NSTIs, urging to broadly identify all present microorganism both in necrotic and macroscopically healthy tissue [20]. Furthermore, ongoing developments of shotgun metagenomics may in the future add genomic information by detecting virulence or resistance determinants that could improve clinical management [35].

## 4. Antibiotic Treatment

As discussed above, prospective or controlled data regarding antibiotic treatment specifically for NSTIs is scarce, with recommended antibiotic regimens mainly based on expert consensus. Some knowledge can however be derived from infections sharing similar characteristics, including non-necrotic skin and soft tissue infections, acute pancreatitis necrosis infection, or conditions with impaired regional perfusion such as diabetic foot or limb ischemia infection.

We will present available data on the choice of molecules and modalities of treatment, before proposing a pragmatic antibiotic treatment scheme for suspected NSTI.

### 4.1. Available Data Regarding Choice of Molecules Specifically in NSTIs

As reported in a meta-analysis of randomized trials on antibiotics in complicated skin and soft-tissue infections [36], there is no evidence to support the use of one antibiotic over others. One open label randomized trial evaluated moxifloxacin *versus* amoxicillin-clavulanate in 804 patients with complicated skin infections, which included 54 NSTIs [37]. There was no difference in clinical success rate, both in the whole population (80.6% vs. 84.5%) or in the subgroup of NSTI patients (50.0% vs. 53.8%). One non-interventional uncontrolled prospective study reported tigecycline use in 163 ICU-admitted patients with complicated skin infections, including 50 NSTI patients [38]. Tigecycline was used as a second-line treatment in 80% of patients, and as a single agent in 64% of cases. The reported cure rate was 90.2%. Likewise, although all were approved for complicated skin and soft tissue infections by non-inferiority trials, there is no evidence for a benefit to recent molecules such as daptomycin, ceftaroline, or ceftobiprole.

The only molecule evaluated in higher numbers of NSTI patients is clindamycin, although the available literature, which is discussed below, mostly included invasive GAS infections overall.

### 4.2. Specificities for GAS Infections: Anti-Toxinic Molecules

Pathogenicity of GAS NSTI seems to largely rely on the bacterial toxins that induce a strong immune response, including polyclonal T-cell activation and cytokine release. Although GAS infections remain highly susceptible to beta-lactams, particularly penicillin, association with clindamycin is strongly recommended in the case of NSTI [11]. Clindamycin is a lincosamine derivative, inhibiting protein synthesis by binding to the 50S subunit of bacterial ribosomes, and retaining bacteriostatic activity on stationary-growth phase bacteria in a mouse model [39]. In line with its mechanism of action, in vitro studies have demonstrated the capacity of clindamycin to significantly inhibit the production of pyrogenic exotoxins SPE-A and SPE-B, also known as superantigenic exotoxins [40,41], as well as other major virulence factors implicated in severe GAS infections [42]. Experimental data from animal models underline the importance of using high doses of clindamycin, as increased protein expression can be observed when subinhibitory clindamycin concentrations are administered [43].

Interestingly, the anti-toxinic benefits of clindamycin were reported regardless of strain susceptibility [43]. Moreover, clindamycin does not appear to be susceptible to inoculum size through—the so-called Eagle effect—as opposed to penicillin [39,44]. In a recent epidemiological study in Spain assessing the resistance profile of invasive *Streptococcus pyogenes* isolates (*n* = 1893), clindamycin resistance was found in only 4.3% of cases [45], but up to 14.6% in the USA [46], and clindamycin should thus not be used as an empirical monotherapy.

Observational studies have suggested a benefit of clindamycin as an adjunctive treatment in patients with invasive GAS (iGAS) infections, including NSTIs, supporting current recommendations by professional societies [11]. Carapetis et al. reported 84 cases of iGAS infection, of whom clindamycin-treated patients had more severe disease than clindamycin-untreated patients but lower mortality at day 30 (15% vs. 39%), with a significantly protective odds ratio for death in univariate but not in multivariate analysis adjusting for the presence of streptococcal toxic shock syndrome and age [47]. Recently, Babiker et al. performed a propensity-matched analysis on a retrospective multicenter cohort including 1079 patients with iGAS infections (including 275 skin and soft tissue infections (SSTIs) but only 12 NSTIs), 343 of whom had received clindamycin adjunct. In the propensity-matched cohort, mortality was significantly lower in patients who received adjunctive clindamycin compared to those who did not (6.5% vs. 11.0%) [48]. Interestingly, such an effect was not confirmed in patients with invasive non-group A/B β-haemolytic streptococcal infections.

Although less documented, linezolid, a member of the oxazolidinone class which also binds the 50S ribosomal subunit, might have comparable in vitro consequences on the toxin release by GAS [49].

### 4.3. Perspectives on Other NSTI Specificities

Recent studies, extensively by the INFECT study group [30], have highlighted other specificities of NSTIs that could impact antibiotic treatment, including the existence of a complex pathobiome in polymicrobial infection [20], of biofilm formation in GAS infections [50], and the intracellular survival of pathogens such as *S. aureus* and GAS [51]. The clinical implications of such findings, i.e., the benefit of using antibiotics particularly active in the setting of biofilm like rifampicin, or with high intracellular concentrations, such as macrolides or tetracyclines, are yet to be determined.

### 4.4. Duration of Treatment and De-Escalation

No study has evaluated the impact of treatment duration on NSTI outcome, resulting in heterogenous practices [52]. Guidelines suggest maintaining treatment for 48–72 h after the last surgery [11], as this delay seems adequate to asses clinical improvement including the absence of fever [53]. Although there is no data specifically on NSTI, antibiotic de-escalation based on microbiological documentation from blood cultures and preoperative samples seems appropriate.

## 5. Pharmacokinetics (PK) and Pharmacodynamics (PD) Targets and Optimization

The mainstem of urgent empirical antibiotic treatment for NSTI is a broad spectrum ß-lactam. These antibiotics display time-dependent activity, where bacterial killing and treatment efficacy correlate with the duration of time that free (unbound) plasma drug concentrations remain above the minimum inhibitory concentration (MIC) of the offending pathogen (fT > MIC) [54]. One major consequence of septic shock, which affects half of NSTI patients [4], is the intense vasodilation and extravasation of fluid into the interstitial space from endothelial damage and capillary leakage. This phenomenon is commonly described as ‘third spacing’. In response to the resulting hypotension, clinicians administer large volumes of resuscitation fluids that may also distribute into interstitial space, thereby significantly increasing interstitial volume. For hydrophilic antibiotics, these processes may lead to a large increase in the distribution volume (Vd), an extensively documented phenomenon in critically ill patients [55].

In addition, hypoalbuminaemia is a common but frequently neglected condition in septic patients. With decreasing albumin concentrations, an increase in the unbound fraction of drugs can occur. The unbound fraction of antibiotics is not only available for elimination, but also for distribution. The Vd for moderate to highly protein-bound antibiotics can thus increase by up to 100% in critically ill patients with hypoalbuminaemia [56,57]. Overall, the Vd of antimicrobials may increase up to 3-fold in critically ill patients, requiring an initial high-loading dose [58,59].

Lastly, severe infections can cause vascular dysfunction, including microvascular failure, which can impair drug delivery into body tissues, particularly in patients with sepsis being treated with vasopressors. In this context, plasma concentrations may profoundly underestimate tissue concentrations [60].

There is evidence defining the impact of early and appropriate antibiotic administration on decreased mortality [61,62]. In sepsis studies, interventions aimed at optimizing antibiotic therapy demonstrate the greatest improvements in clinical outcomes [63,64]. Convincing in vitro, animal and clinical data also suggest that microbiological success may be optimal when serum β-lactam concentrations are maintained above 4 × MIC during the entire time frame [65,66,67,68,69]. The results of the DALI study indeed support the conclusions that better outcomes for critically ill patients can be expected with higher drug exposures [63]. Prolonged infusion schemes increase the fraction of the dose interval in which unbound antibiotic concentrations exceed the MIC of the pathogen (fT > MIC), as compared with standard intermittent infusion [64,70,71,72,73,74]. A meta-analysis showed that hospital mortality was significantly lower (RR, 0.74; 95% CI, 0.56–1.00; *p* = 0.045) and clinical cure significantly higher (RR, 1.20; 95% CI, 1.03–1.40; *p* = 0.021) in the continuous infusion group than in the intermittent infusion group in severe patients [75]. In a meta-analysis of randomized trials, the risk of death in patients with sepsis treated with prolonged infusion of anti-pseudomonal β-lactams was 30% lower compared with patients treated with short term infusion [76], a finding confirmed in another meta-analysis focusing on piperacillin-tazobactam [77]. However, the stability over time of the drug solution may challenge the continuous infusion of some antibiotics (e.g., imipenem or amoxicillin-clavulanate) [78]. Despite high doses of β-lactams and continuous administration, desirable PK/PD targets (i.e., >4 MIC) may not be achieved in patients with augmented creatinine clearance [66]. Thus, given intra- and/or inter-individual PK variability and defined targets, experts recommend therapeutic drug monitoring in critically ill adult patients [79].

## 6. Tissue Diffusion of Antibiotics

Other features of NSTIs raise the interest of optimizing antibiotic tissue delivery, as despite urgent and extensive removal of necrotic tissues as a mainstem of management [3,11,12], a median of 2 to 4 reinterventions are needed [4,7]. Local drug delivery may not only be impaired by global circulatory failure and the above discussed pharmacokinetic alterations induced by septic shock, but also by tissue necrosis and altered regional tissue perfusion from local micro-vessel dysfunction and thrombosis [60] (Figure 1). Available data approaching such conditions are derived from non-necrotizing SSTI, diabetic foot infection, acute limb ischemia, or acute pancreatitis-related necrosis infection. Only one recent study included 11 obese patients with severe SSTI, of whom 9 had NSTI, and evaluated the pharmacokinetics of linezolid. Although thought to have an excellent soft tissue distribution, the probability of target attainment for this drug was low using the standard dosing of 600 mg every 12 h, highlighting the importance of conducting specific studies in this specific population [80].

### 6.1. Available Data from Uncomplicated SSTI

Soft tissue distribution is reportedly high for lipophilic molecules, such as oxazolidinones, quinolones, macrolides, metronidazole, daptomycin and tigecycline [81,82,83,84,85]. Although, it was reported to be satisfactory for aminoglycosides [86]. Though it could be lower for beta-lactams [81,87], particularly in the case of high-protein binding [88], small scale pharmacokinetic studies on healthy subjects or monte-carlo simulations have found prolonged time periods above the minimum inhibitory concentrations 90 (MIC_90_) of *S. aureus* and *S. pyogenes* in blisters for meropenem, imipenem and piperacillin-tazobactam [89,90], and high probabilities of target attainment for cefazolin in the interstitial fluid [91]. One recent study suggested higher probabilities of target attainment for ceftaroline than for vancomycin, linezolid or daptomycin, but it must be noted that this study was conducted by the firm licensing ceftaroline, and pooled data from literature not exclusively focusing on SSTI [92]. Takesue et al. found no correlation between PK/PD parameters and clinical success for daptomycin in uncomplicated *S. aureus* SSTIs [93]. Overall, data in non-necrotic SSTIs are reassuring regarding the use of beta-lactams for empiric therapy. Nevertheless, in the setting of severe SSTIs, such as NSTIs, optimizing PK/PD for such hydrophilic drugs by using high doses and continuous infusion could improve outcome and seems reasonable [81] (Table 2).

**Table 2 antibiotics-10-01104-t002:** Pharmacokinetics and pharmacodynamics characteristics of the main molecules recommended for the empiric treatment of NSTIs in the main international and national guidelines.

Molecule	Guidelines	Pharmacokinetic Parameters			Tissue Penetration (Tissue/Blood Ratio)			Antimicrobial Spectrum/Anti-Toxinic Activity and other Specific Aspects	Dosing Regimen ^§^
		Distribution (Vd)	Protein Binding	PK/PD Targets in Severe Infections	Soft Tissue of Healthy Subjects	Impact of Necrosis on Tissue Diffusion	Impact of Altered Tissue Perfusion on Tissue Diffusion [84,94,95,96,97,98]		
Piperacillin + tazobactam	USA 2014 (IDSA) World 2018 (WSES/SIS-E) Germany 2018 South Korea 2017	Hydrophilic (0.24 L/kg)	Low (16%)	fT > 4−8 × MIC = 100%	Low (0.27) [81] *	Medium to low impact (tissue concentrations appear sufficient to achieve PK/PD targets)	No data	Methicillin-susceptible *S. aureus, S. pyogenes, Enterobacteriaceae*, nonfermenting bacilli, anaerobic bacteria	4 g q6h IV Consider prolonged (4 h) or continuous infusion with loading dose
Cefotaxime	USA 2014 Norway 2013 South Korea 2017	Hydrophilic (0.28 L/kg)	Low (30–51%)	fT > 4−8 × MIC = 100%	Medium (0.54) [99] ^π^	Medium to low impact (tissue concentrations appear sufficient to achieve PK/PD targets)	No data (but high decrease of tissue concentration with others cephalosporins: cefepime [100] and ceftazidime [101])	Methicillin-susceptible *S. aureus, S. pyogenes, Enterobacteriaceae*	2 g q6–8h IV
Meropenem	USA 2014 World 2018 Germany 2018 South Korea 2017	Hydrophilic (0.25 L/kg)	Very low (2%)	fT > 4−8 × MIC = 100%	Low (0.35–0.48) [87] *	Medium to high decrease in drug concentration (tissue concentrations could not be sufficient to achieve PK/PD targets)	Low impact	Methicillin-susceptible *S. aureus, S. pyogenes, Enterobacteriaceae*, nonfermenting bacilli anaerobic bacteria activity on multi-drug resistant gram-negative bacilli	1–2 g q8h IV Consider prolonged infusion 3 h
Gentamycin	Norway 2013	Hydrophilic (0.26 L/kg)	Very low (0–3%)	C_max_/MIC > 8–10	Medium (0.60) [81] *	High decrease in drug concentration	No data	*S. aureus, S. pyogenes, Enterobacteriaceae,* nonfermenting bacilli Rapid bactericidal action Should be added in cases of septic shock	5–8 mg/kg over 30 min, q24h
Amikacin	France 2018	Hydrophilic (0.26 L/kg)	Very low (< 10%)	C_max_/MIC > 8–10	High (1.03) [86] ^π^	High decrease in drug concentration	No data	*S. aureus, S. pyogenes, Enterobacteriaceae,* nonfermenting bacilli Rapid bactericidal action Should be added in cases of septic shock	25–30 m/kg over 30 min, q24h
Metronidazole	USA 2014 Norway 2013 South Korea 2017	Lipophilic (0.65 L/kg)	Very low (< 10%)	AUC_24_/MIC C_max_/MIC No target defined	High (0.67) [85] *	No or low impact	No impact [102]	Anaerobic bacteria	500 mg q8h IV
Vancomycin	USA 2014 South Korea 2017	Hydrophilic (0.70 L/kg)	Medium (55%)	AUC_24_/MIC > 400–600	Low (0.30) [81] *	Medium to high decrease in drug concentration	High impact [103]	Methicillin-resistant *S. aureus*	Consider continuous infusion of 30 mg/kg/24 h with loading dose of 30 mg/kg and TDM
Daptomycin	World 2018	Hydrophilic (0.10 L/kg)	High (92%)	AUC_24_/MIC > 666 [104]	High (0.74–0,93) [81] *	No data	No impact	Methicillin-resistant *S. aureus*	8–12 mg/kg q24h
Linezolid	USA 2014 World 2018 South Korea 2017	Lipophilic (0.65 L/kg)	Low (31%)	AUC_24_/MIC > 80–120 fT > 1 × MIC = 85% [105]	High (0.75–1.32) [81] *	No data	No impact	Methicillin-resistant *S. aureus* In vitro evidence of anti-toxinic action	600 mg q12h IV (higher doses might be needed in obese patients [80])
Clindamycin	World 2018 Germany 2018	Lipophilic (1.1 L/kg)	High (90%)	AUC_24_/MIC No target defined	High (1.06) [106] ^x^	No data	No impact	*S. aureus*, *S. pyogenes* Anaerobic bacteria (but with high proportion of resistant strains), High evidence of in vivo and in vitro anti-toxinic action	600–900 mg q8h IV

* Microdialysis technique. ^π^ Skin blister fluid technique. ^x^ Tissue digestion technique. ^§^ These dosages are defined for an adult of standard weight, without renal or hepatic impairment. Dosage adjustment rules should be applied as appropriate. AUC: Area under curve. Cmax: Maximal concentration. fT: fraction of time. IDSA: Infectious Diseases Society of America. IV: Intra venous. MIC: Minimal inhibitory concentration. NSTI: Necrotizing soft tissue infection. PD: Pharmacodynamics. PK: Pharmacokinetics. SIS-E: Surgical Infection Society of Europe. Vd: Distribution Volume. WSES: World Society of Emergency Surgery. The choice of molecules depends on the local ecology and individual risk factors for carrying resistant bacteria (e.g., ESBL and MRSA). Vancomycin, linezolid, or daptomycin should be added to a beta-lactam regimen in case of identified risk factors for MRSA infections. Clindamycin should be added in case of limb NSTI for its good anti-toxinic activity but should not be used as empirical monotherapy. Aminoglycosides should be added in case of septic shock. Carbapenems should be added in case of risk factors for drug-resistant gram-negative bacilli (see Figure 2 legend).

**Figure 2 antibiotics-10-01104-f002:**
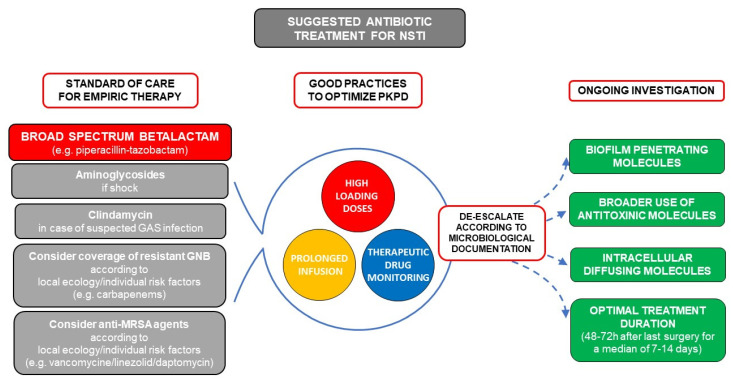
Suggested antibiotic treatment for necrotizing soft tissue infection (NSTI) and future perspectives. The mainstem of empiric treatment is a broad-spectrum beta-lactam (e.g., piperacillin-tazobactam) with additional aminoglycosides in case of septic shock. Clindamycin should be added in case of documented or suspected group A streptococcus (GAS) infection (limb infection, features of streptococcal toxic shock, absence of comorbidities, blunt trauma, absence of chronic skin lesions, homelessness, injectable drug use, non-steroidal anti-inflammatory drug use). Coverage of resistant gram-negative bacilli by carbapenems should be used according to local ecology and individual risk factors (hospital acquired infection, beta-lactam, or quinolone exposure in the previous 3 months, history of extended spectrum beta-lactamase (ESBL) carrying, germ colonization/infection or travel to high ESBL endemicity aeras in the previous 3 months). Similarly, use of anti-methicillin resistant *Staphylococcus aureus* (MRSA) drugs such as vancomycin, linezolid, or daptomycin should be considered in case of local endemicity, residence in a long-stay care facility, chronic dialysis, permanent transcutaneous medical devices or prior MRSA infection/colonization. Pharmacokinetics (PK) and Pharmacodynamics (PD) should be optimized by use of high-loading doses and prolonged infusions for molecules with time-dependent bactericidal activity such as beta-lactams, and therapeutic drug monitoring should be used when available.

### 6.2. Available Data on Antibiotic Diffusion in Necrotic Tissue

These are derived from infection of necrosis in acute pancreatitis [107,108]. Although these are somewhat dated studies which did not use dosing regimens and administration modalities that are now the standard of care, they highlight the impact of molecule choice in tissue diffusion. Indeed, although the tissue/serum concentration ratio was high and tissue concentrations above the MICs of identified microorganisms were found for quinolones and metronidazole in the vast majority of patients, this was not always the case for beta-lactams, such as imipenem, and never the case for aminoglycosides. Tissue penetration was also suggested to be dependent on the degree of necrosis, inflammation, and regional perfusion [109]. In a recent literature review, piperacillin-tazobactam or cefepime diffusion into pancreatic tissue seemed superior to that of meropenem, abounding for the use of carbapenem only when resistant pathogens are suspected [110]. Although similar profiles are expected in skin and soft tissue necrosis, studies to evaluate the impact of molecule choice and administration modalities on tissue diffusion and outcome are needed in NSTI.

### 6.3. Available Data on Antibiotic Diffusion in the Presence of Altered Perfusion

Data is scarce regarding this issue, mainly derived from acute limb ischemia and diabetic foot infection. Interestingly, skin exudate antibiotic concentrations after intravenous administration were proposed as a means to evaluate local circulatory conditions, highlighting the link between tissue perfusion and drug delivery [111]. Experimental data from animal models have shown that even after ischemia has resolved, drug delivery can be hindered for as long as 72 h [112]. Lozano-Alonso et al. prospectively evaluated soft tissue and bone diffusion of antibiotics among 61 patients with acute limb ischemia requiring amputation [94]. Drug tissue concentration depended on tissue perfusion for vancomycin, levofloxacin, and ceftazidime, but not meropenem, clindamycin, or linezolid. Although diffusion of clindamycin is allegedly good in skin and soft tissue [43,113] and was not affected by tissue perfusion, it did not reach adequate concentrations for the least sensitive strains, such as anaerobes of the *Bacteroidetes* genre, in this study. Others found no impact of ischemia on diffusion of metronidazole [102]. In diabetic foot infection, a condition known to be associated with altered local perfusion, lipophilic molecules such as daptomycin, linezolid, and trimethoprim-sulfamethoxazole have shown good tissue penetration [95,96,97,98], while it was altered for hydrophilic molecules such as vancomycin [103]. While ceftazidime penetration was not affected by diabetes, it highly depended on tissue perfusion in another work [101], and employing high doses is suggested for cefepime [100]. Maximal concentration in soft tissue was both reduced and delayed when compared to healthy-volunteers for tedizolid and ceftolozane-tazobactam [114,115]. Interestingly, tissue diffusion of linezolid in two patients was reported to improve after hyperbaric oxygen therapy, a therapeutic intervention sometimes employed in NSTIs [116].

Overall, data on antibiotic skin and soft tissue penetration in the setting of necrosis and altered perfusion abounds for improving PK/PD for hydrophilic molecules such as beta-lactams. Administration modalities should definitely be optimized with high-loading doses initially, continuous perfusion and therapeutic drug monitoring when available, as this has shown improved tissular penetration in other settings [77,117,118], but also in skin and soft-tissue infections, for example for piperacillin-tazobactam [88]. Whether the use of lipophilic drugs, less sensitive to such alterations, with higher reported tissue diffusion than beta-lactams such as clindamycin, quinolone, metronidazole, linezolid, daptomycin or tetracyclines has a beneficial impact on outcomes remains however unsettled. Β-lactams, combined with clindamycin when there is a high probability of GAS infection, remain the mainstem of antibiotic treatment in NSTIs until such data are available.

There are only anecdotical reports on the use of topical antibiotics in NSTIs [119], as was suggested in necrotic pancreatitis [120]. Although their added benefit to parenteral treatment and surgery was not investigated, this approach does not seem suited as an exclusive first-line therapy.

## 7. Suggested Empiric Treatment for Suspected NSTI Based on Basic Microbiology

The treatment of NSTIs relies on antibiotics and early surgical debridement, which is one of the most important modifiable prognostic factors [8]. With more than 50% of patients presenting with septic shock, urgent and bactericidal intravenous antibiotics are recommended [11,12]. NSTIs are often polymicrobial, and although some admission characteristics have been correlated with monomicrobial forms [18,121,122], even the site of infection is insufficient to guide empiric antibiotic treatment [4,19,123,124]. This should cover both gram positive, gram-negative, and anaerobic bacteria, usually with a broad-spectrum β-lactam (e.g., piperacillin-tazobactam). As discussed above, high-loading doses, continuous infusion, and therapeutic drug monitoring could improve the outcome. According to local ecology and individual risk factors, coverage of MRSA by glycopeptides or daptomycin, or of resistant gram-negatives by carbapenems should be considered on a case-by-case basis. Aminoglycosides should be reserved to broadening spectrum in case of septic shock. Finally, clindamycin adjunction seems adequate in the case of proven or suspected GAS infection (limb infection, features of streptococcal toxic shock, absence of comorbidities, blunt trauma, absence of chronic skin lesions, homelessness, injectable drug use, and non-steroidal anti-inflammatory drug use). In the absence of data, de-escalation of spectrum according to documentation seems reasonable, and suggested treatment duration is of 48–72 h after last surgery in case of clinical improvement [11,12]. A suggestion for management of antibiotic treatment in NSTI as well as future perspectives is presented in Figure 2.

## 8. Conclusions

Together with urgent surgical removal of necrotic tissues, antibiotics are the cornerstone of NSTI management, its mainstem being urgent bactericidal intravenous administration of a broad-spectrum beta-lactam such as piperacillin-tazobactam. Though no literature specifically focusing on NSTI exists, clinicians face the association of the profound PK/PD alterations associated with sepsis (i.e., distribution volume increase, tissue necrosis, locally altered perfusion) hindering drug diffusion. Administration modalities should definitely be optimized with high initial loading doses, continuous antibiotic perfusions and therapeutic drug monitoring when available. Future research focusing on antibiotic strategies in NSTI patients should assess the use of drugs with higher reported tissue diffusion, broader use of anti-toxinic, biofilm, or intracellular penetrating molecules as well as the effect of different/personalized treatment durations.

## Figures and Tables

**Figure 1 antibiotics-10-01104-f001:**
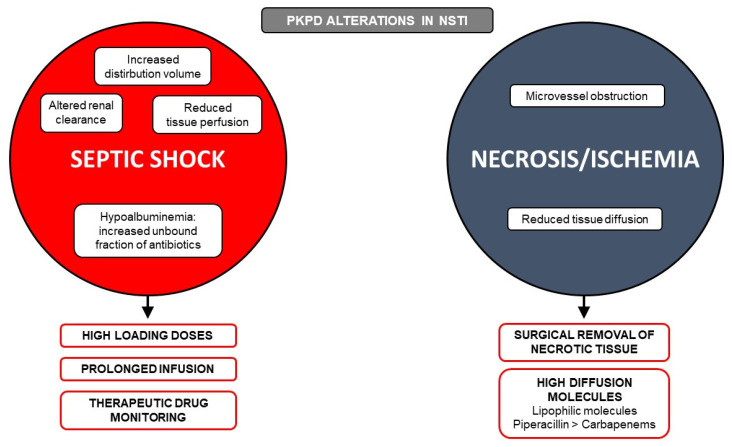
Pharmacokinetic (PK) and pharmacodynamic (PD) alterations in patients with necrotizing soft tissue infection (NSTI). Features of septic shock from any cause (increased distribution volume, altered renal clearance, hypoalbuminemia, and reduced tissue perfusion) abound for optimizing delivery of hydrophilic and time-dependent drugs such as beta-lactams by using high-loading doses and prolonged infusion with therapeutic drug monitoring. Specificities of NSTI with tissue necrosis and local ischemia resulting in hindered tissular diffusion are consistent with the need for urgent and aggressive surgical debridement of necrotic tissues. Molecules with higher tissue diffusion, such as lipophilic molecules (e.g., clindamycin, linezolid and daptomycin), might be of interest in this setting.

**Table 1 antibiotics-10-01104-t001:** Most frequently cultured microorganisms in NSTI. Classification adapted from Bruun et al. [25].

Type of Pathogens	Primary	Secondary	Polymicrobial	Commensals
Pathogenicity	May cause NSTI in patients without known risk factors	May cause infection in patients with risk factors	Rarely pathogen in the absence of a primary or secondary pathogen	Do not cause NSTI although sometimes identified with other pathogens
Species	Group A *Streptococcus* *Staphylococcus aureus* *Vibrio vulnificus* *Clostridium perfringens*	Other *Streptococcus* (group B, C, G, *anginosus*) *Pneumococcus* *Haemophilus influenzae* *Neisseria meningitidis* *Enterobacteriaceae* Nonfermenting gram-negative bacilli Other anaerobes (*Bacteroides*, *Prevotella*, *Fusobacterium*)	*Enterococcus*	*Bacillus* *Corynebacterium* *Micrococcus* Coagulase negative *Staphylococci*

## Data Availability

Not applicable.

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
