# Peer review of "Antibiotics in Necrotizing Soft Tissue Infections"

_antibiotics, 2021, doi:10.3390/antibiotics10091104_

Round 1

Reviewer 1 Report

Urbina and colleagues have provided a comprehensive and detailed discussion of the use of antibiotics during the treatment for necrotizing soft tissue infections (NSTI).  They have provided numerous relevant references and discuss differing circumstances that might affect antibiotic administration.  There is only one area that I believe needs correction. 

In Table 1 they list potential primary microorganisms that can be responsible as sole pathogens for NSTI and list Group A Streptococcus, Vibrio vulnificus, and Clostridium perfringens.  They do not list Staphylococcus aureus in this column, and it should be listed as well.  One of their own references (Huang TY, Peng KT, Hsiao CT, Fann WC, Tsai YH, Li YY, et al. Predictors for gram-negative monomicrobial necrotizing fasciitis in southern Taiwan. BMC infectious diseases. 2020;20(1):60.) lists single organism NSTI episodes and reports more Staph aureus cases than Streptococcal cases.  Other references on this topic not included in their bibliography are

Miller LG, Perdreau-Remington F, Rieg G, Mehdi S, Perlroth J, Bayer AS, et al. Necrotizing fasciitis caused by community-associated methicillin-resistant Staphylococcus aureus in Los Angeles. The New England journal of medicine. 2005;352(14):1445-53,

which reports a series of NSTI cases caused by community acquired MRSA with Panton-Valentine leucocidin,

Lee TC, Carrick MM, Scott BG, Hodges JC, Pham HQ. Incidence and clinical characteristics of methicillin-resistant Staphylococcus aureus necrotizing fasciitis in a large urban hospital. Am J Surg. 2007;194(6):809-12; discussion 12-3,

that reports a series of NSTI cases, 39% caused by MRSA, 

Wong CH, Tan SH, Kurup A, Tan AB. Recurrent necrotizing fasciitis caused by methicillin-resistant Staphylococcus aureus. Eur J Clin Microbiol Infect Dis. 2004;23(12):909-11. Epub 2004 Nov 13.

describing a case caused by MRSA without any prior trauma, and

Cheng NC, Wang JT, Chang SC, Tai HC, Tang YB. Necrotizing fasciitis caused by Staphylococcus aureus: the emergence of methicillin-resistant strains. Ann Plast Surg. 2011;67(6):632-6. doi: 10.1097/SAP.0b013e31820b372b,

which reports 17% of their NSTI caused by Staph, and

Shumba P, Mairpady Shambat S, Siemens N. The Role of Streptococcal and Staphylococcal Exotoxins and Proteases in Human Necrotizing Soft Tissue Infections. Toxins. 2019;11(6),

a review paper on NSTI which reports GAS and Staph aureus is the two major pathogens associated with monomicrobial NSTIs.

Author Response

Urbina and colleagues have provided a comprehensive and detailed discussion of the use of antibiotics during the treatment for necrotizing soft tissue infections (NSTI).  They have provided numerous relevant references and discuss differing circumstances that might affect antibiotic administration. There is only one area that I believe needs correction. 

Authors : we would like to thank this reviewer for appraising our manuscript and for the thoughtful and constructive comments raised. 

In Table 1 they list potential primary microorganisms that can be responsible as sole pathogens for NSTI and list Group A Streptococcus, Vibrio vulnificus, and Clostridium perfringens. They do not list Staphylococcus aureus in this column, and it should be listed as well.  One of their own references (Huang TY, Peng KT, Hsiao CT, Fann WC, Tsai YH, Li YY, et al. Predictors for gram-negative monomicrobial necrotizing fasciitis in southern Taiwan. BMC infectious diseases. 2020;20(1):60.) lists single organism NSTI episodes and reports more Staph aureus cases than Streptococcal cases. 

Authors : We agree with the reviewer. Table 1 has been modified as suggested and now includes « Staphylococcus aureus ».

Other references on this topic not included in their bibliography are

Miller LG, Perdreau-Remington F, Rieg G, Mehdi S, Perlroth J, Bayer AS, et al. Necrotizing fasciitis caused by community-associated methicillin-resistant Staphylococcus aureus in Los Angeles. The New England journal of medicine. 2005;352(14):1445-53,

which reports a series of NSTI cases caused by community acquired MRSA with Panton-Valentine leucocidin,

Lee TC, Carrick MM, Scott BG, Hodges JC, Pham HQ. Incidence and clinical characteristics of methicillin-resistant Staphylococcus aureus necrotizing fasciitis in a large urban hospital. Am J Surg. 2007;194(6):809-12; discussion 12-3,

that reports a series of NSTI cases, 39% caused by MRSA, 

Authors : We thank the reviewer for these suggestions. The two references cited above have now been added to the revised version of the manuscript (references 22 and 23 of the revised text) and the following sentences have been added :

« In some series, S. aureus was reported to be more prevalent than GAS in monomicrobial NSTIs [18]. Community-associated methicillin-resistant S. aureus carrying the Panton-Valentine leucocidin have been reported in the United States [22,23]. »

Wong CH, Tan SH, Kurup A, Tan AB. Recurrent necrotizing fasciitis caused by methicillin-resistant Staphylococcus aureus. Eur J Clin Microbiol Infect Dis. 2004;23(12):909-11. Epub 2004 Nov 13.

describing a case caused by MRSA without any prior trauma, and

Cheng NC, Wang JT, Chang SC, Tai HC, Tang YB. Necrotizing fasciitis caused by Staphylococcus aureus: the emergence of methicillin-resistant strains. Ann Plast Surg. 2011;67(6):632-6. doi: 10.1097/SAP.0b013e31820b372b,

which reports 17% of their NSTI caused by Staph, and

Shumba P, Mairpady Shambat S, Siemens N. The Role of Streptococcal and Staphylococcal Exotoxins and Proteases in Human Necrotizing Soft Tissue Infections. Toxins. 2019;11(6),

a review paper on NSTI which reports GAS and Staph aureus is the two major pathogens associated with monomicrobial NSTIs.

Authors : We thank the reviewer for these suggestions. We have not added these references because : the reference by Wong et al. is a single case report, the article by Cheng et al. included a small number of patients (n=18) and we believe the conclusions drawn by this article are limited, and the article by Shumaba et al. is a review article not including original data.

Reviewer 2 Report

I read with great interest the paper. I find it well wrote and the question research is very relevant

Below my suggestions

  • implement better the section on etiopathogenesis and causes of necrotizing fasciitis
  • add data on polymicrobial aspects
  • discuss the outcome of necrotizing fascitis
  • Table 2 is very very excellent. Very good job
  • Discuss better the role of antibiotic diffusion in the presence of altered perfusion and the best therapeutic choice
  • Underline the coordinate action surgery + antibiotic therapy that need this complicate diseases

Congratulations to the authors for the article 

Author Response

I read with great interest the paper. I find it well wrote and the question research is very relevant

Authors : we would like to thank this reviewer for appraising our manuscript and for the constructive comments raised, which certainly helped improving the quality of the manuscript. 

Below my suggestions

  • implement better the section on etiopathogenesis and causes of necrotizing fasciitis

Authors : Because the aim of the current article was to review the current knowledge regarding antibiotic use in NSTIs and because the article is already 3701 words long, we have not written an extensive etiopathogenesis section. Yet, as suggested by the reviewer, we have added the following sentences to the revised « Introduction » and « Microbiology of NSTIs » sections, so that to highlight some crucial points pertaining to the risk factors and pathophysiology of NSTIs :

« Most patients developing NSTIs have previous comorbidities, including diabetes mellitus, obesity, cardiovascular disease, intravenous drug use and immunosuppression [1–3]. Infection can spread after traumatic injuries, minor breaches of the skin or mucosa, and even non penetrating soft tissue injuries [1]. »

« A recent study has highlighted that streptococcal NSTIs were associated with different host responses than polymicrobial NSTIs, with higher expression of interferon-inducible mediators and lower expression of extracellular matrix components [19]. »

  • add data on polymicrobial aspects

Authors : the following sentences have been added to the revised « Microbiology of NSTIs » sections

« Anaerobic, aerobic and facultative anaerobic bacterials act synergistically and fuel a cycle of bacterial colonization and inflammatory tissue necrosis [19]. »

« A recent study has highlighted that streptococcal NSTIs were associated with different host responses than polymicrobial NSTIs, with higher expression of interferon-inducible mediators and lower expression of extracellular matrix components [19]. »

We have also mentioned the categorization of poly- and monomicrobial NSTIs in type I and type II infections as it has been used in several previous studies.

  • discuss the outcome of necrotizing fascitis

Authors : As mentioned above, the main objective of the current article is not to deeply present data regarding the outcome of NSTIs. The Introduction section of the manuscript already provides general information regarding outcome :

« Mortality ranges from 10 to 30% according to initial patient severity, and morbidity among survivors includes potential amputations and profound impact on long-term health-related quality of life [5–7].

Because the article is already 3701 words long and because it is beyond the scope of the present review to extensively review the outcome of NSTIs, we would prefer not to expand the informations on this aspect.

  • Table 2 is very very excellent. Very good job

Authors : We thank the reviewer for their positive comment.

  • Discuss better the role of antibiotic diffusion in the presence of altered perfusion and the best therapeutic choice

Authors : a whole section (section 6.3 « Available data on antibiotic diffusion in the presence of altered perfusion ») is dedicated to this aspect and contains 463 words. The following sentences had already been written regarding « best therapeutic choices »:

« Overall, data on antibiotic skin and soft tissue penetration in the setting of necrosis and altered perfusion abounds for improving PK/PD for hydrophilic molecules such as beta-lactams. Administration modalities should definitely be optimized with high loading doses initially, continuous perfusion and therapeutic drug monitoring when available, as this has shown improved tissular penetration in other settings [77,117,118], but also in skin and soft-tissue infections, for example for piperacillin-tazobactam [88]. Whether the use of lipophilic drugs, less sensitive to such alterations, with higher reported tissue diffusion than beta-lactams such as clindamycin, quinolone, metronidazole, linezolid, daptomycin or tetracyclines has a beneficial impact on outcomes remains however unsettled. »

 In order to adress the reviewer’s comment and provide the reader with pragmatic informations, we have added the following sentence to finalize this paragraph :

« Β-lactams, combined with clindamycin when there is a high probability of GAS infection, remain the mainstem of antibiotic treatment in NSTIs until such data are available. »

  • Underline the coordinate action surgery + antibiotic therapy that need this complicate diseases

 Authors : We do agree with the point raised by the reviewer. However, this crucial aspect has already been underlined in the introduction section (« Initial urgent management of NSTIs relies on broad-spectrum antibiotic therapy, rapid surgical debridement of all infected tissues and, when present, treatment of associated organ failures in the intensive care unit. Time to surgery is one of the main modifiable prognostic factors, with, in a recent meta-analysis, a significantly lower mortality rate when surgery was performed within six hours of hospital admission [8].”) and conclusion (“Together with urgent surgical removal of necrotic tissues, antibiotics are the cornerstone of NSTI management “) of the article.

To further highlight this point, the following sentence has been added to the revised section 7 (“Suggested empiric treatment for suspected NSTI based on basic microbiology”):

The treatment of NSTIs relies on antibiotics and early surgical debridement, which is one of the most important modifiable prognostic factors [8] »

Congratulations to the authors for the article 

Authors : we thank the reviewer for their positive comment on our work.